# Modification of the Surface of 40 Kh Steel by Electrolytic Plasma Hardening

**Zhuldyz Sagdoldina** [1,2], **Laila Zhurerova** [1], **Yuri Tyurin** [3], **Daryn Baizhan** [1,2], **Aizhan Kuykabayeba** [4], **Saule Abildinova** [5] **and Rauan Kozhanova** [2,6,*]

1   Research Center "Surface Engineering and Tribology", Sarsen Amanzholov East Kazakhstan University, Ust-Kamenogorsk 070000, Kazakhstan
2   Research Center "Material Surface Modification", Shakarim University, Semey 071412, Kazakhstan
3   E.O. Paton Electric Welding Institute, National Academy of Sciences of Ukraine, 03650 Kyiv, Ukraine
4   Department of Thermal Physics and Technical Physics, Al Farabi Kazakh National University, Almaty 050040, Kazakhstan
5   Department of Management and Entrepreneurship in Engineering, Almaty University of Power Engineering and Telecommunications, Almaty 050013, Kazakhstan
6   PlasmaScience LLP, Ust-Kamenogorsk 070010, Kazakhstan
*   Correspondence: kozhanovars@yandex.kz; Tel.: +7-775-668-62-39

**Abstract:** The high-strength, medium-carbon alloy construction steel 40 Kh is commonly used in the manufacture of tools and machine parts. This paper experimentally investigates the effect of electrolytic plasma thermocyclic hardening on the surface hardening and microstructure modification of 40 Kh steel. The research was carried out using optical microscopy, scanning electron microscopy, X-ray diffraction analysis and micro-hardness measurements. Modified samples were obtained at different electrolyte plasma thermal cycling modes. As a result of the heat treatment, hardened layer segments of different thicknesses and structural composition formed on the surface of the steel. The parameters and mechanisms of surface hardening were determined by examining the microstructural modification and phase transformation both before and after treatment. It was revealed that the main morphological structural-phase component of the initial state of 40 Kh steel was a ferrite–pearlite structure, and after electrolytic plasma thermocyclic hardening, the hardened martensite phase was formed. It was found that in order to achieve a hardening depth of 1.6 mm and an increase in hardness to 966 HV, the optimum time for electrolytic plasma treatment of 40 Kh steel was 2 s. The technology under discussion gives an insight into the surface hardening potential for improving the service life and reliability of 40 Kh steel.

**Keywords:** electrolytic plasma thermocyclic hardening; structure; hardness; phase; martensite





## 1. Introduction

The surface finishing of machine parts and tools is carried out to extend the life and performance of metal components as, in most cases, the surface layers determine the quality level of the material [1,2]. The surface hardening of tools and machine parts is heated by radiation from process lasers, electron guns or high frequency currents [3–7]. However, the high cost of equipment and low efficiency in the use of material and energy resources limit the use of these technologies.

Traditionally, only the heating and cooling of steel has been the most common surface treatment method that has existed over time. Conventional heat treatment consists of heating the metal in a furnace and cooling in air, water or oil [8]. All of these processes require an expensive set-up and longer processing times, and some have cumbersome equipment [9]. Electrolytic plasma surface hardening (heating–hardening) is a new method that overcomes all these disadvantages [10]. This proves that it can successfully improve

the desired physical and mechanical properties in much less time; in about a few seconds, compared to traditional heat treatment processes which require hours and days. Electrolytic plasma hardening (heating–hardening) is a special thermomechanical process which uses an electrolyte in an aqueous solution under certain conditions, e.g., voltage, current, electrolyte, duration and speed of heating–hardening [11,12]. In the heating–plasma hardening process, electrical energy is normally transferred from the metal anode to the workpiece itself through layers of electrolyte and plasma. The plasma layer is formed from the electrolyte material in the gap between the liquid electrode and the conductive surface of the workpiece. Electrolytic plasma hardening is a complex process that combines physical metallurgy and electrochemical processes, such as cathodic heating of the sample, where phase transformation and deformation occur simultaneously. Electrolytic plasma hardening of medium-carbon steels can alter their microstructure, causing changes in the mechanical and physical properties and affecting their behavior under operating conditions [13–17]. One of the important parameters determining the nature of phase change during electrolyte plasma heating is the periodic increase in temperature at which the metal is heated above the phase transition temperature $\alpha \rightarrow \gamma$. Subsequent rapid cooling captures the iron-based phases formed in the machined material layer as martensite, bainite and fine pearlite, respectively, and has industrially valuable material properties. The transition layer comprises: a fine-grained structure characteristic of the heat-affected zone and a coarse-grained structure characteristic of the substrate [18]. Our previous studies [19,20] have comparatively studied the effect of volume and surface heat treatments on the structural-phase states of medium carbon steel. Surface hardening was carried out by electrolyte plasma method. Volumetric hardening of the samples was carried out by heating to 900 °C with subsequent cooling in water and oil, and a part of the samples after hardening was annealed at 510 °C. It was found that electrolyte plasma surface hardening leads to an increase in microhardness up to two times due to the formation of fine needle martensite. This proves that the electrolyte plasma hardening method can be considered as an alternative to the traditional method of heating and cooling in the furnace.

Cathodic electrolytic plasma hardening allows a wide variation of the heating and cooling rates (50–500 °C/s) and produces a treated layer with a thickness of 0.1 to 10 mm [21]. The authors of [22] presented the results of electrolyte plasma thermal cyclic treatment of AISI 4140 steel. According to the results, the treatment process was carried out at cyclic voltages of 320 and 250 V with a time of 1, 2, 3, 4 s and 2, 3, 4, 5 s, respectively. It was determined that the hardened layer consisted of martensite phases, fine-dispersed pearlite and iron oxides. Depending on the conditions of electrolyte plasma thermocycling, the depth of the hardened layer was 6–8 mm and the corresponding microhardness was equal to 500–650 HV, which is 4–5 times higher than that of the matrix. Therefore, the results obtained by these authors also confirm that the electrolyte plasma thermocyclic treatment can change the structural-phase state, as well as improve the physical and mechanical properties that positively affect the service life of medium-carbon steels under operating conditions [23–25].

In addition, because plasma formation is instantaneous and takes place within a few seconds, it becomes difficult to stabilize the process input parameters to produce output parameters that suit our needs. Due to existing technical problems, this process is still under development [26,27]. To improve the structure of steels and their mechanical properties, we recommend the development of an electrolytic plasma thermocyclic hardening (EPTCH) mode based on the application of cyclic thermal influences.

In view of the above, the aim of the present work is to study the effect of an electrolytic plasma thermocyclic hardening mode on the structure and hardness of medium carbon steel.

## 2. Materials and Methods

Test specimens were made of 40 Kh steel of the following chemical composition: 0.36% C; 0.5% Mn; 0.2% Si; 0.8% Cr; 0.3% Ni; 0.3% Cu; 0.035% S; 0.035% P. Before experimenting with the steel surface, all $2 \times 2 \times 1$ cm$^3$ specimens were hand-sanded on abrasive paper with grit sizes ranging from P100 to P2000, polished on velvet and nylon cloths with 0.25–0.5 μm diamond paste and then cleaned with alcohol. EPTCH of steel was carried out in the cathode mode at the electrolyte-plasma treatment plant, the scheme of which is shown in Figure 1b. The power source was a powerful rectifier giving a maximum output of 360 V/100 A as a direct current [28,29]. A sodium carbonate ($Na_2CO_3$) solution was used as a heating and cooling source. The electrolyte composition was 85% distilled water and 15% sodium carbonate (weight %). The voltages (U, V) and treatment times (t, s) were different for each sample. The samples were hardened at 320 V, 250 V and 50 V sequences. Periodic changes in the electric field strength between the surfaces of the liquid electrode and the product were found to change the power density of surface heating, which provides control over the electrolyte plasma heating and ensures that the necessary thermal conditions are provided for the formation of hardening structures. The electrolytic plasma pulse values, treatment time and thermal cycling are shown in Table 1.

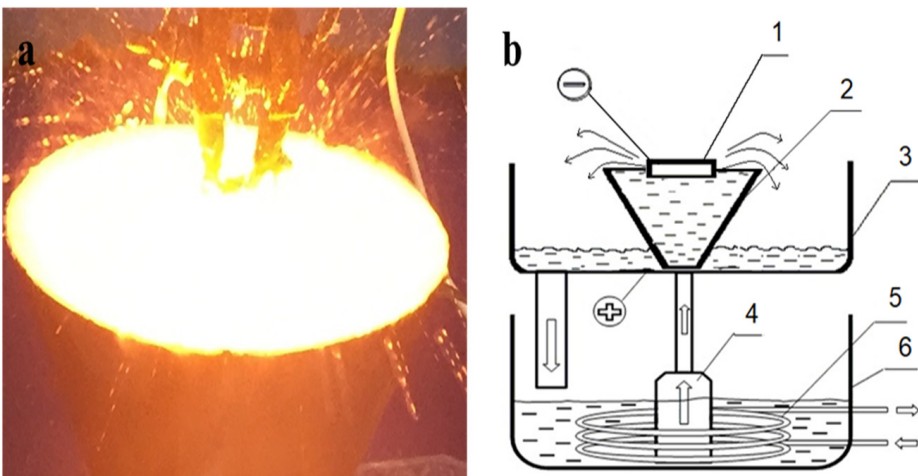

**Figure 1.** (**a**) Electrolyte plasma sample treatment process; (**b**) functional diagram: 1—workpiece; 2—cone-shaped stainless steel electrolytic cell; 3—sump; 4—pump; 5—heat exchanger; 6—electrolyte bath.

**Table 1.** Technological parameters of electrolytic plasma thermal cycling treatment modes for 40 Kh steel.

| Cycle | 1 | | 2 | | 3 | | 4 | | 5 | | 6 | | 7 | |
|---|---|---|---|---|---|---|---|---|---|---|---|---|---|---|
| Sample | $U_1$, V | $t_1$, s | $U_2$, V | $t_2$, s | $U_3$, V | $t_3$, s | $U_4$, V | $t_4$, s | $U_5$, V | $t_5$, s | $U_6$, V | $t_6$, s | $U_7$, V | $t_7$, s |
| No. 1 | 320 | 1 | 50 | 7 | 320 | 1 | - | - | - | - | - | - | - | - |
| No. 2 | 320 | 2 | - | - | - | - | - | - | - | - | - | - | - | - |
| No. 3 | 320 | 1 | 250 | 3 | 50 | 5 | 320 | 1 | 250 | 2 | 50 | 5 | 320 | 1 |

Experimental investigations were carried out on the basis of S.Amanzholov East Kazakhstan University in the research center "Surface engineering and tribology" (Ust-Kamenogorsk, Kazakhstan). The phase composition of the samples was investigated by X'PertPro X-ray diffractometer (Philips, Almelo, the Netherlands) using CuKα radiation. Data processing and quantification were carried out using PowderCell 2.4 (Kraus W, Nolze G., 2000, PowderCell for Windows, version 2.4). The etching was carried out with a 4.0% nitric acid solution ($HNO_3$) in alcohol for 10 s [26]. Optical micrographs were obtained with

an Altami 5C microscope (Altami Ltd., Saint-Petersburg, Russia). The scanning electron microscope (SEM) (MIRA3 LMU, TESCAN, Brno, Czech Republic) was used to study the structure at ×4000 and ×10,000. The chemical composition of the different phases of the microstructure was also studied using an energy dispersive spectrometer (EDS) connected to the SEM, using point scanning techniques. The microhardness of the samples was measured according to GOST 9450-76 (ASTM E384-11) on the microtesting device, Metolab 502 (Metolab, Saint-Petersburg, Russia), at loads on an indenter P = 1 N and exposure time of 10 s [30].

One of the main methods for experimental investigation of the mechanical properties of micro- and nanocomposite materials is the indentation method. Thus far, various types of continuous nanoindentation tests have been increasingly used in practice to measure the mechanical properties of materials, due to the high accuracy with which elastic characteristics are measured. The indentation method determines the hardness, Young's modulus and elastic recovery of both superhard and soft materials using small loads. Nanoindentation analysis was carried out to evaluate the mechanical properties on a micro-scale of hardened 40 Kh steel samples. The modulus of elasticity is determined by the method of indentation on a complex measuring system according to the Oliver–Farr method [31]. It allows us to distinguish two components from the information contained in the unloading branch of the force–deformation diagram, which is related to both the elastic part of the achieved deformation and energy and the plastic one. The Young's modulus E is determined by the measured elastic stiffness of the contact S = $\partial P/\partial h$ at the unloading stage from the following expression:

$$E_r = \frac{\sqrt{\pi}}{2} \frac{S}{\sqrt{A}}, \tag{1}$$

where $E_r = [(1 - \nu_s{}^2)/E_s + (1 - \nu_i{}^2)/E_i] - 1$ is the reduced Young's module, taking into account the individual Young modules E and Poisson coefficients $\nu$ of the contacting bodies; the sample (with index s) and the indenter (with index i); and $A_c$ is the contact area. Nanohardness *H* is defined as:

$$H = \frac{P_{max}}{A} \tag{2}$$

Nanoindentation tests were performed on a tester (Nanoscan 4D compact, Moscow, Russia) using a Berkovich indenter tip. The conducted peak load was 200 mN. The loading/unloading rate was 20 mN/s and was maintained at the maximum load for 10 s. The hardness and Young's modulus were measured in accordance with GOST R 8.748-2011 (ISO 14577-1:2015) [32]. The tribological characteristics of steel were measured in the sliding friction mode according to the "ball-on-disk" scheme on an Anton Paar TRB[3] tribometer (CSM Instruments, Bern, Switzerland). The sample rotation speed was 2 cm/s, the load was 6 N and a ball of $Si_3N_4$ (silicon nitride) with a diameter of 6 mm was used as a counterbody. The total sliding distance reached 60 m, which is enough for the wear testing of the hardened layer.

## 3. Results and Discussion

The microstructures prior to the electroplasma hardening of 40 Kh steel are shown in Figure 2a. As shown in the optical micrographs, the main phase component of the initial state of 40 Kh steel is ferrite (F-white part) and pearlite (P-dark part) [33,34]. After electrolytic plasma thermocyclic hardening, hardened martensite (M) structures formed on the steel surface, which are shown in Figure 2b–d.

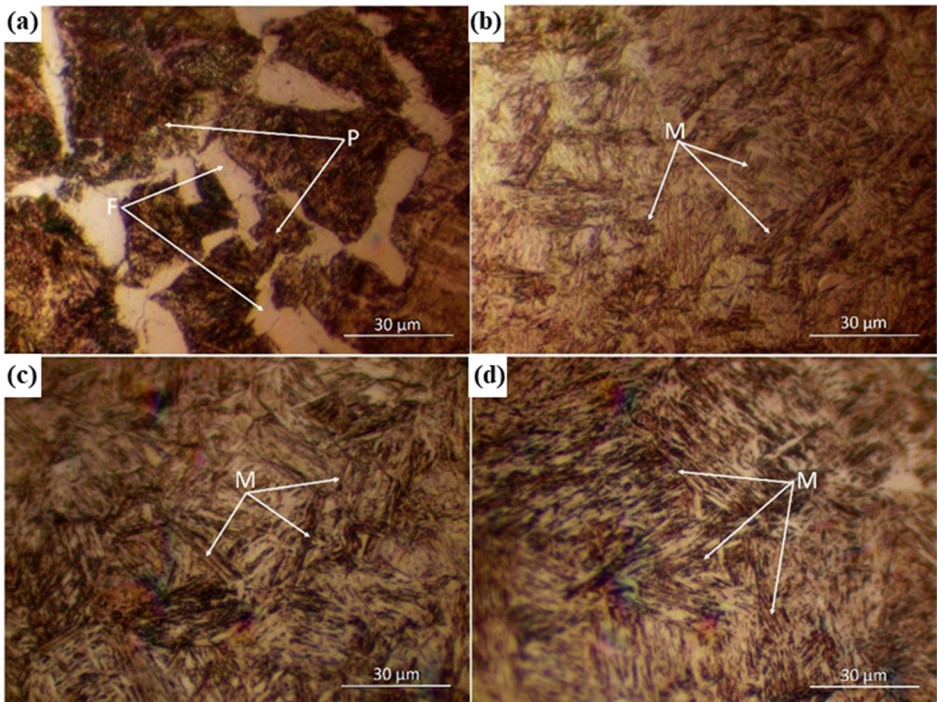

**Figure 2.** Optical micrographs of 40 Kh steel: (**a**) initial, (**b**) No. 1, (**c**) No. 2 and (**d**) No. 3.

Figure 3 shows the diffractograms of the original and hardened 40 Kh steel specimens which were evaluated by X-ray diffraction patterns using the software database, PowderCell 2.4. Diffractograms were taken from the active surface of the original and plasma hardened samples. The results show the presence of peaks corresponding to martensite. These results are in agreement with the microstructure study.

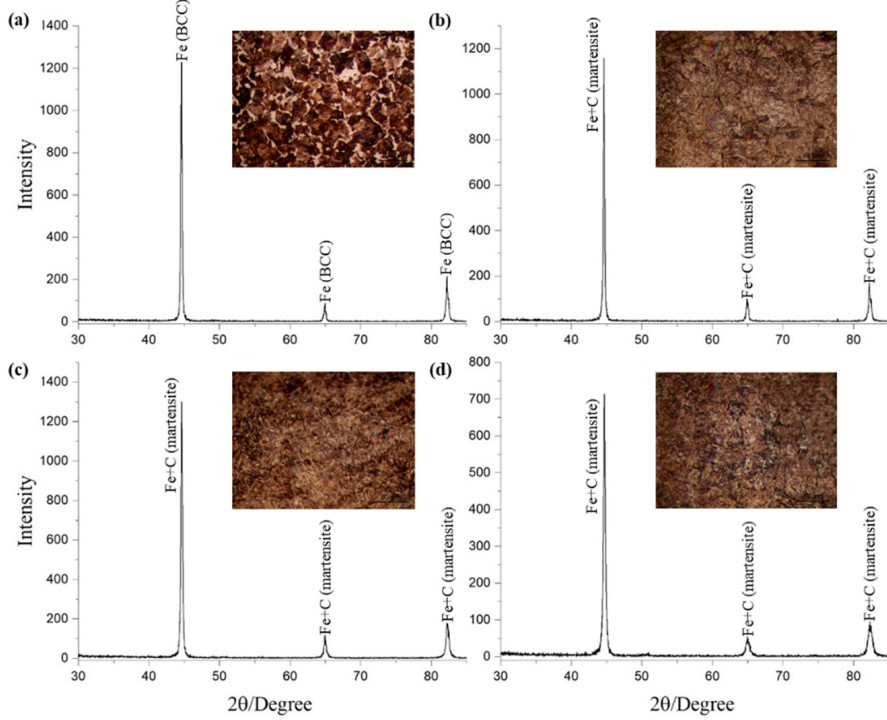

**Figure 3.** Diffractograms of 40 Kh steel: (**a**) initial, (**b**) No. 1, (**c**) No. 2 and (**d**) No. 3.

By periodically switching on a high voltage electrical potential of 320 V and a low voltage (pause) of 50, 250 V, the heating rate was increased and decreased, which then allowed more time and a thicker heated layer. During the demonstrated voltage mode, the pulse mode heated the samples, while the pause mode in the experiment either cooled the samples or maintained a constant temperature. The intermittent coupling of a high (320 V) and a low (pause 50, 250 V) electric potential results in a periodic increase and stabilizes or slows down the increase in heating rate, which then allows more time and a thicker heated layer without melting the surface [35,36]. As shown in Table 2, the electrolyte plasma treatment modes influenced the thickness of the hardened layer. The layer thickness was measured cross-sectionally using an optical microscope. The thickest layer (4.7 mm) was obtained in sample No. 3. For the remaining samples, the thickness of the hardened layer was approximately the same.

**Table 2.** The influence of EPTCH modes on the formation of the thickness of the hardened layer.

| Mode | Cycle | Thickness of the Hardened Layer, mm |
|---|---|---|
| No. 1 | 3 | 1.1 |
| No. 2 | 1 | 1.6 |
| No. 3 | 7 | 4.7 |

Figure 4 shows the Vickers micro-hardness of the samples in different states. In the initial state, 40 Kh steel had a microhardness of 245 HV, and after electroplasma hardening, the highest microhardness was at sample No. 2 which was 920 HV. The reason for the change in microhardness is the periodic increase in temperature at which the steel is heated above the phase transition temperature $\alpha \rightarrow \gamma$ and the austenite to martensite is converted when cooling after the exposure. This transformation occurs in all hardened samples.

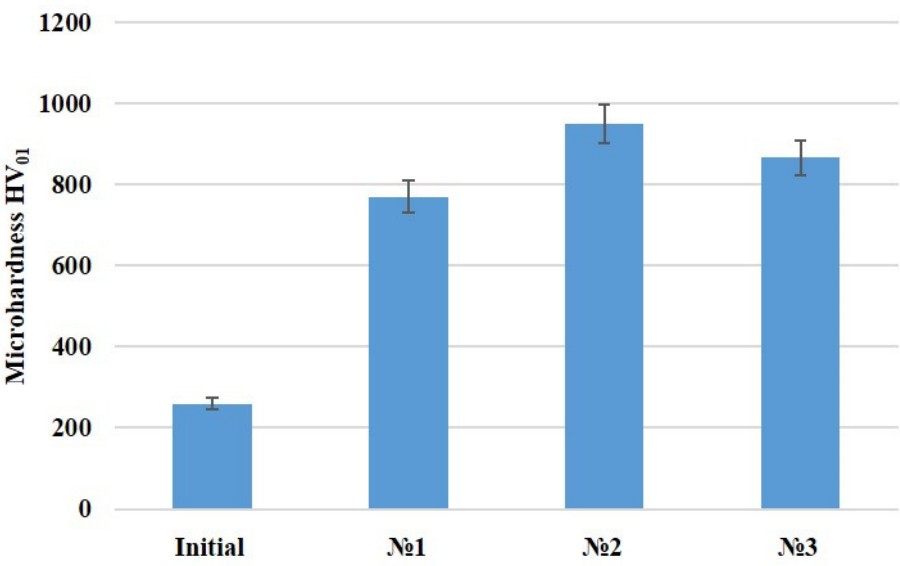

**Figure 4.** Vickers micro-hardness of the samples in different states.

The nanoindentation method is widely used to measure the mechanical properties of materials at the microscale. Nanoindentation is a modern field for determining the hardness of materials, which is very useful in the field of mechanical engineering. A typical experimental curve for this method in the form of a graph of the dependence of the load (P) on the depth of indentation (h) is shown in Figure 5. The curve consists of two parts corresponding to the process of loading and unloading.

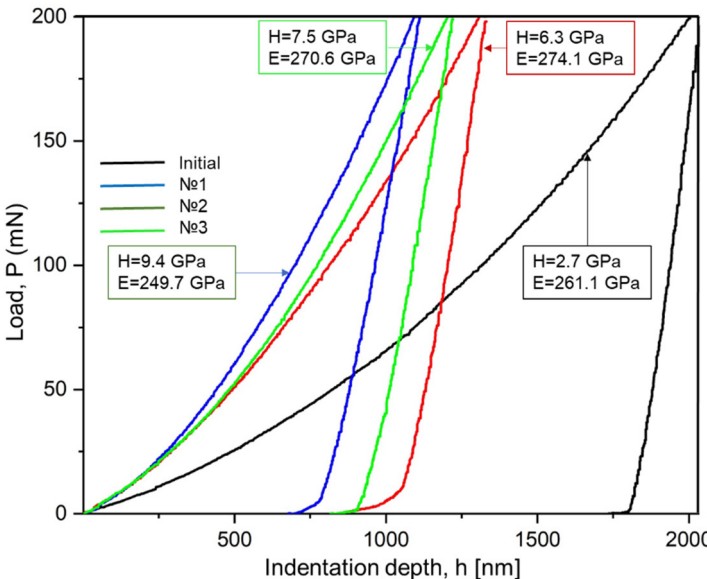

**Figure 5.** Load–displacement curves for steel 40 Kh before and after EPTCH.

According to the data, EPTCH nanoindentation has a very strong effect on the hardness of 40 Kh steel. The EPTCH process transformed the martensitic phase. The formation of martensite led to an improvement in hardness compared to the original steel. These results show that EPTCH treatment contributes to the increase in the hardness of the samples due to the formation of the hardening phases, like martensite.

It is known from the wear theory that the structural state of steels also significantly affects the wear resistance [37]. As a rule, a martensitic structure demonstrates better wear resistance compared to a ferritic, pearlitic and bainitic structure [38]. To determine the tribological parameters of the surface, tribological tests for the sliding friction of the surfaces were carried out on a ball-on-disk scheme on an automated friction machine, Tribometer TRB[3] [39]. The distribution graph of the sliding friction coefficient and the wear volume of 40 Kh steel at different thermal cycling modes are shown in Figure 6. It was found that the higher hardness of the EPTCH samples in the current experiment significantly limited the depth of deformation of the worn surface, which led to a significant decrease in friction coefficients. By comparing the values of the friction coefficient and wear volume of the original steel with a ferrite–pearlite structure with the hardened steel samples with a martensitic structure, it can be stated that hardness is certainly an important property of wear-resistant materials. Because the low hardness of the sample surface causes deeper penetration of abrasive particles, it therefore makes the micro-cutting process more efficient in addition to the more pronounced deformation of the material.

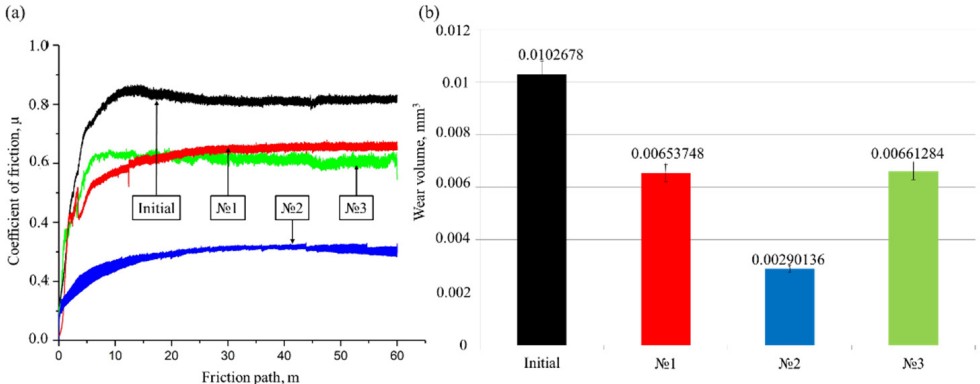

**Figure 6.** Distribution graph of (**a**) coefficient of sliding friction and (**b**) wear volume of 40 Kh steel.

In connection with the above generalized results of the EPTCH of 40 Kh steel under different thermal cycling modes, it was customary to consider sample No. 2 as the optimal mode, which has an improved structural-phase and mechanical characteristics in many respects. Figure 7 shows a cross-section of specimen No. 2 consisting of three zones comprising the base metal, the heat-affected zone and the hardening zone. The results of the 40 Kh microstructure in the hardening zone consist mainly of martensite, while bainite and highly dispersed pearlite are formed in the transition zone. After electrolytic plasma hardening (heating–hardening), the microstructure changes according to the Fe-C diagram of state [40,41]. Layers are formed in which the structural constituents are present according to the carbon content of the steel when quenched from temperatures above $AC_1$ and cooled below it, with transition zones between the layers and with structural constituents of adjacent areas.

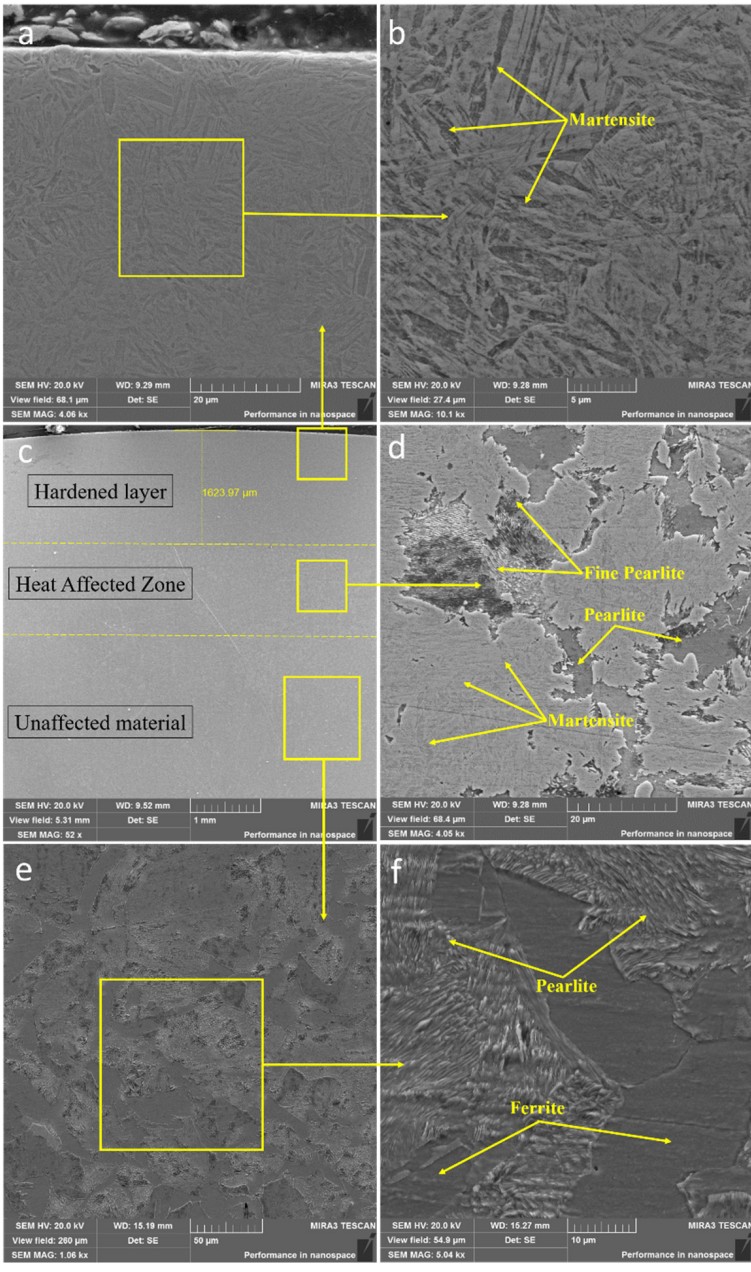

**Figure 7.** SEM images of the cross-section of sample No. 2. (**a**) hardened layer at ×4000 and (**b**) ×10,000, (**c**) micrography at ×50, (**d**) heat affected zone at ×4000, (**e**) unaffected material at ×1000 and (**f**) ×5000.

To experimentally verify these claims, special investigations were carried out to determine the chemical composition of steel using an energy dispersive spectrometer (EDS) connected to an SEM, using point scanning techniques [42,43]. SEM micrographs and the corresponding EDS spectra of the steel samples are illustrated in Figure 8. The EDS results verify the formation of the $\alpha'$ phase by indicating 0.7 wt% C in this phase as illustrated in Figure 8a. The chemical analysis data for the depth of electroplasma hardening and the unstrengthened zone, presented in Table 3, confirm the chemical microheterogeneity of the structural and phase components of the investigated steel.

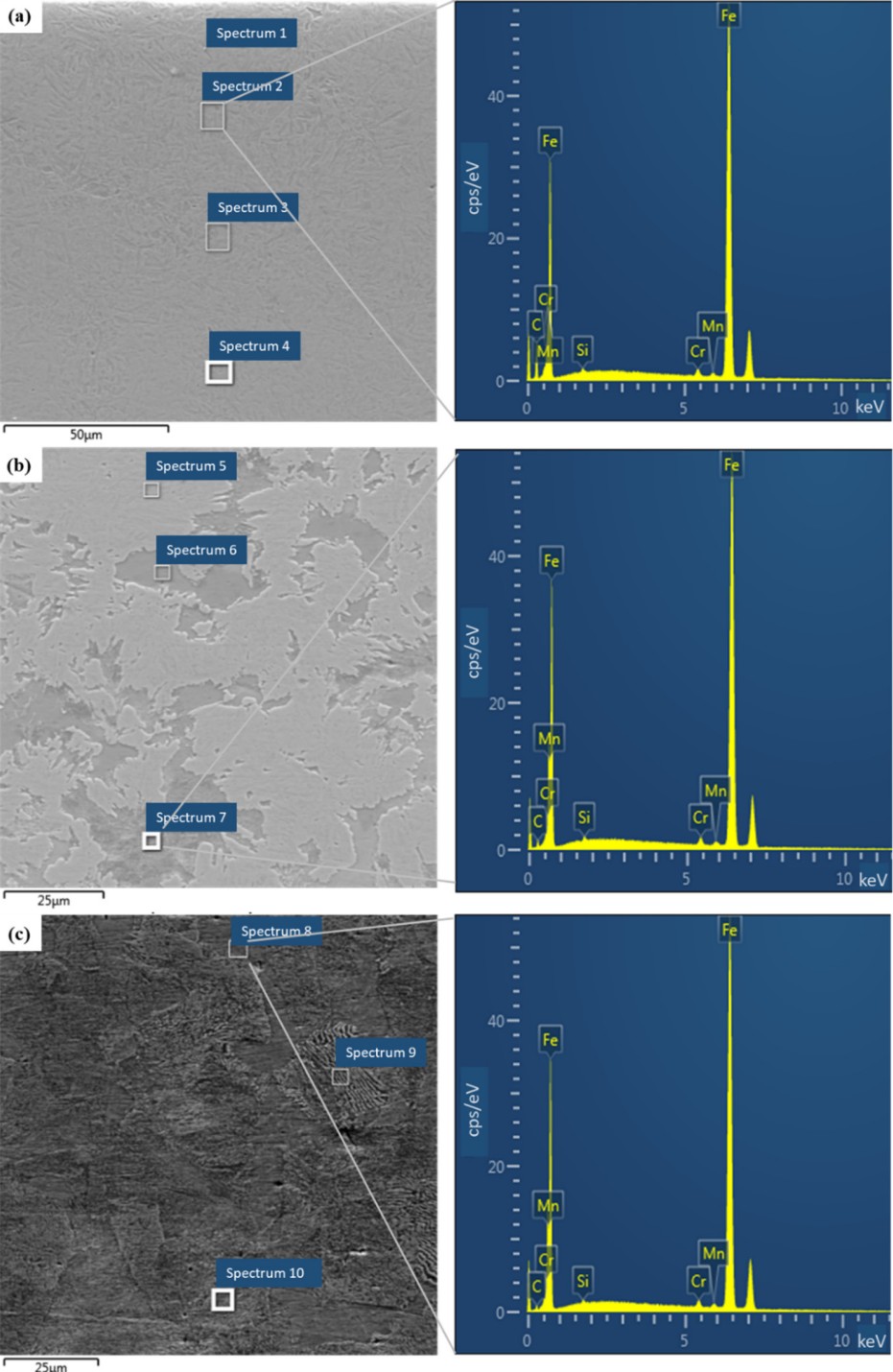

**Figure 8.** Schematic graph of EDS testing zones: (**a**) hardened layer, (**b**) heat affected zone and (**c**) unaffected material.

**Table 3.** EDS analysis results.

| Spectrum | Fe (Weight %) | Cr (Weight %) | Mn (Weight %) | Si (Weight %) | C (Weight %) |
|---|---|---|---|---|---|
| Spectrum 1 | 97.4 | 1.0 | 0.6 | 0.3 | 0.7 |
| Spectrum 2 | 97.4 | 1.0 | 0.6 | 0.4 | 0.7 |
| Spectrum 3 | 97.3 | 0.9 | 0.8 | 0.3 | 0.6 |
| Spectrum 4 | 97.3 | 1.1 | 0.7 | 0.2 | 0.7 |
| Spectrum 5 | 97.7 | 0.9 | 0.7 | 0.3 | 0.4 |
| Spectrum 6 | 97.6 | 1.0 | 0.6 | 0.3 | 0.5 |
| Spectrum 7 | 97.5 | 1.1 | 0.6 | 0.3 | 0.4 |
| Spectrum 8 | 97.7 | 1.0 | 0.7 | 0.3 | 0.3 |
| Spectrum 9 | 97.8 | 0.9 | 0.7 | 0.3 | 0.3 |
| Spectrum 10 | 97.5 | 1.1 | 0.8 | 0.3 | 0.3 |

Figure 9 shows the microhardness values along the depth of the steel. The hardness of the hardened layer is within 950 HV. The high heating and cooling rates ensure that hardening structures with a smooth transition to the base metal are formed (Figure 7). The thickness of the transition layer is 0.5–0.6 mm. At a depth of 1.1 mm, a martensitic structure with microhardness 971–864 HV is observed, at a depth of 1.2–1.6 mm, a highly dispersed pearlitic–bainitic structure with microhardness 685–445 HV is observed, followed by a base metal structure with a typical ferrite–pearlite structure with microhardness 260–263 HV. The creation of a system of hard inclusions on the surface of products provides multiple increases in wear resistance and compressive stresses in the surface layer of the product, which increase its strength [44,45].

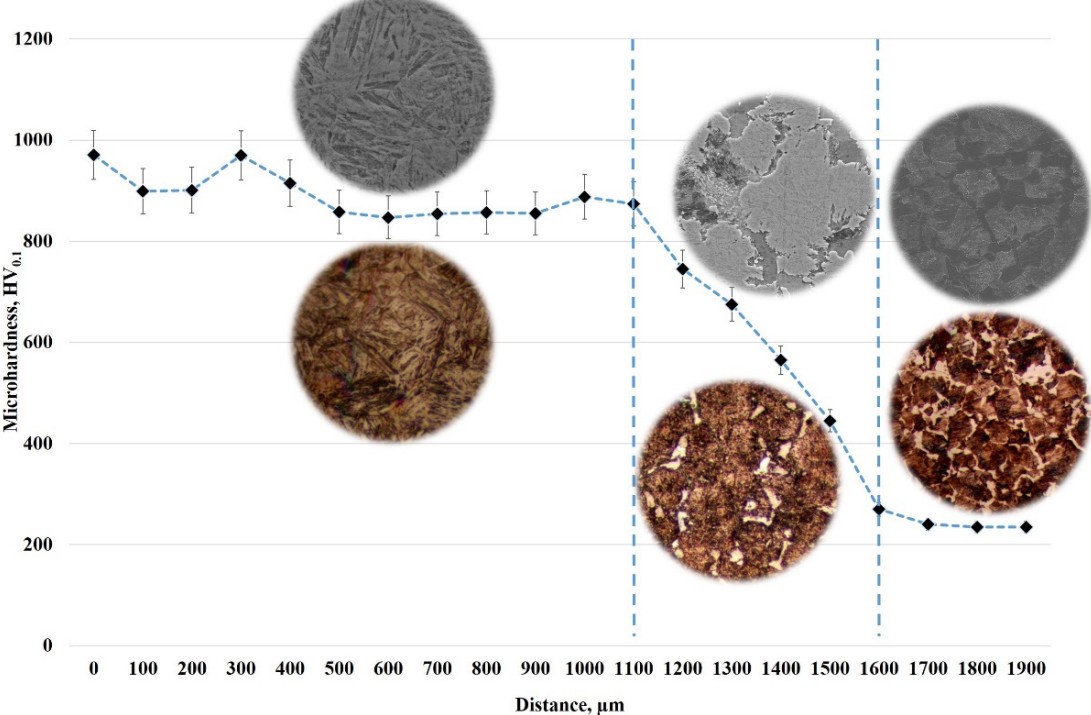

**Figure 9.** Depth distribution of microhardness in sample No. 2 after EPTCH.

## 4. Conclusions

Electrolytic plasma hardening (heating–hardening) of the surface of 40 Kh steel produces positive results with a carbon content of more than 0.3% and sufficient alloying elements to produce solid phases. The intermittent coupling of a high (320 V) and low (pause 50, 250 V) electric potential leads to a periodic increase and stabilizes or delays the heating rate increase, resulting in a significant increase in hardening depth (1.1–4.7 mm) and an increase in maximum hardness in microvolumes to 770–965 HV in the 40 Kh steel surface layer. The results show that electrolytic plasma hardening (heating–hardening) has a high potential in controlling major structural changes in the surface layer of steel in the formation of physical and mechanical properties.

**Author Contributions:** Conceptualization, Y.T. and S.A.; methodology, D.B. and Y.T.; investigation, L.Z., A.K. and Z.S.; writing—original draft preparation, R.K.; visualization, L.Z. and Y.T.; writing—review and editing, L.Z. All authors have read and agreed to the published version of the manuscript.

**Funding:** This research has been funded by the Science Committee of the Ministry of Education and Science of the Republic of Kazakhstan (Grant No. AP08857733).

**Data Availability Statement:** Not applicable.

**Conflicts of Interest:** The authors declare no conflict of interest.

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
