# Peer review of "Modification of the Surface of 40 Kh Steel by Electrolytic Plasma Hardening"

_metals, doi:10.3390/met12122071_

Round 1
Reviewer 1 Report
The manuscript is interesting and well written.
I only recommentd to make the scale bar of the micrographs more uniform and to increase the font size of some axes labels (for example, the ones in Figure 8).
Author Response
We thank the reviewer for his in-depth analysis of our manuscript and his comments and suggestions, which helped us to significantly improve the quality of our data analysis and presentation. In the revised version of this paper, we have tried, as much as possible, to take into account the comments and suggestions presented in the review.
Below are the responses to the comments
Comments for Author:
- I only recommentd to make the scale bar of the micrographs more uniform and to increase the font size of some axes labels (for example, the ones in Figure 8).
Response: We have made changes to Figure 2 and made the scale ruler uniform. The font size in Figure 8 has been increased.

Reviewer 2 Report
This paper systematically studied the effect of electrolytic-plasma thermocyclic hardening on the surface hardening and microstructure modification of 40Kh steel. It has been revealed that the main morphological structural-phase component of the initial state of steel 40Kh is ferrite-perlite structure, and after electrolytic-plasma thermocyclic hardening g the hardened martensite phase is formed. It was found that in order to achieve a hardening depth of 1.6 mm and an increase in hardness to 966 HV the optimum time for electrolytic plasma treatment of 40Kh steel is 2 seconds.
Here are some comments for the authors:
1. All SEM pictures should be added professional ruler, the rulers on the raw SEM pictures are not clear enough.
2. Figure 5, Which sample does the blue curve correspond to?
3. Why you choose 40Kh steel?
4. After EPTCH, the hardness of the sample is increased, which is not enough to prove that its service life would be improved, wear resistance evaluation is suggested to be added?
Author Response
We thank the reviewer for his in-depth analysis of our manuscript and his comments and suggestions, which helped us to significantly improve the quality of our data analysis and presentation. In the revised version of this paper, we have tried, as much as possible, to take into account the comments and suggestions presented in the review.
Below are the responses to the comments
Comments for Author:
- All SEM pictures should be added professional ruler, the rulers on the raw SEM pictures are not clear enough.
Response: We have made a change in Figures 7 and 9.
- Figure 5, Which sample does the blue curve correspond to?
Response: We agree with the reviewer. In Figure 5, the blue curve corresponds to sample No. 2.
- Why you choose 40Kh steel?
Response: 40Kh steel is widely used for the manufacture of critical components of the automotive industry, such as parts of internal combustion engines, gears and various parts of complex configurations, which require high surface hardness and wear resistance.
- After EPTCH, the hardness of the sample is increased, which is not enough to prove that its service life would be improved, wear resistance evaluation is suggested to be added?
Response: We agree with the reviewer. A change has been made to the article. To determine the tribological parameters of the surface, tribological tests were performed on the sliding friction of surfaces according to the “ball-on-disk” scheme using an auto-mated friction machine Tribometer TRB3. The graph of the distribution of the sliding friction coefficient of steel 40kh under different thermal cycling modes is shown in Figure 6.

Reviewer 3 Report
The paper proposed an idea to improve surface hardness of 40Kh specimen by using an electrolytic-plasma thermocyclic hardening method. It is valuable for the colleagues to reference. The writing of the paper is clear for readers to understand. However, the quality of the paper could be improved for exhibiting the importance of the research work.
The design of the experiment could be improved logistically. It could be suggested to explain how to choose the experimental parameters, for example, electrolytic-plasma pulse values, treatment time and thermal cycling in table 1..
The figure 2 should be more clear to identify the phases.
The units used in the paper should be changed to international standard if necessary.
The detail of the measurement about the thickness of hardness layer and a picture showing the hardness layer should be provided.
Author Response
We thank the reviewer for his in-depth analysis of our manuscript and his comments and suggestions, which helped us to significantly improve the quality of our data analysis and presentation. In the revised version of this paper, we have tried, as much as possible, to take into account the comments and suggestions presented in the review.
Below are the responses to the comments
Comments for Author:
- The design of the experiment could be improved logistically. It could be suggested to explain how to choose the experimental parameters, for example, electrolytic-plasma pulse values, treatment time and thermal cycling in table 1.
Response: We agree with the reviewer. Studies have shown that when a 320 V electric potential heater is connected to the electric circuit, the sample surface is heated. The power density of the energy at the cathode is so high that almost 5–10 sec later the melting of the surface layer begins. The heating rate on the surface of the product reaches 500 °C/s.
- The figure 2 should be more clear to identify the phases.
Response: We agree with the reviewer. We have made changes to Figure 2.
- The units used in the paper should be changed to international standard if necessary.
Response: We agree with the reviewer. A change has been made to the article.
- The detail of the measurement about the thickness of hardness layer and a picture showing the hardness layer should be provided.
Response: We agree with the reviewer. A change has been made to the article. Below are the drawings of the hardened layer and hardness.
Hardness and microstructure of the cross-section of sample No. 2.

Reviewer 4 Report
1. How electrolytic-plasma hardening (heating-hardening) has a high potential in controlling major structural changes?
2. Test specimens - standards followed in testing is to be eloborated?
3. Give more details on Nanoindentation tests.
4. How it is ensured the high heating and cooling provide a smooth transition to the base metal?
5. Literature review is poor. Include the follwoing literatures in the revision
(a) https://doi.org/10.12982/CMJS.2022.034;
(b) DOI 10.1088/2053-1591/abf3e7
6. English grammar and typos errors are to be checked
Author Response
We thank the reviewer for his in-depth analysis of our manuscript and his comments and suggestions, which helped us to significantly improve the quality of our data analysis and presentation. In the revised version of this paper, we have tried, as much as possible, to take into account the comments and suggestions presented in the review.
Below are the responses to the comments
Comments for Author:
- How electrolytic-plasma hardening (heating-hardening) has a high potential in controlling major structural changes?
Response: We agree with the reviewer. Electrolyte-plasma thermocyclic surface hardening is an attractive solution for heat treatment used to improve the properties of the steel surface by structural and phase transformation. Structural and phase transformations during electrolyte-plasma thermocyclic hardening are performed repeatedly at varying heating–cooling temperatures, which radically improve the quality of the part. The effectiveness of the effect of electrolytic-plasma thermocyclic treatment on the structure of steel is largely determined by the mode of its implementation, that is, the temperatures in the cycle, the number of cycles, as well as the rate of heating and cooling.
- Test specimens - standards followed in testing is to be eloborated?
Response: We agree with the reviewer. A change has been made to the article.
- Give more details on Nanoindentation tests.
Response: We agree with the reviewer. A change has been made to the article.
- How it is ensured the high heating and cooling provide a smooth transition to the base metal?
Response: We agree with the reviewer. A change has been made to the article. The heating of the metal depends on the applied voltage and the duration of the process, or we can say on the temperature. The thickness of the hardened layer depends on this. Cooling takes place in the electrolyte. For example, on sample No. 2, we heated it to a temperature of 880-900 C and immediately cooled it to room temperature in the electrolyte. Heated to a temperature within the austenitic phase region and the sample was completely austenized, then cooled to room temperature in the electrolyte. The final microstructure is martensite. Both SEM and optical microscope drawings show a strong formation of martensite.
- Literature review is poor. Include the follwoing literatures in the revision
(a) https://doi.org/10.12982/CMJS.2022.034;
(b) DOI 10.1088/2053-1591/abf3e7
Response: We agree with the reviewer. A change has been made to the article.
- English grammar and typos errors are to be checked.
Response: We agree with the reviewer. A change has been made to the article.

Round 2
Reviewer 2 Report
Authors have sufficiently revised the manuscript, and it is suggested to be accepted.
Author Response
We thank the reviewer for his in-depth analysis of our manuscript and his comments and suggestions, which helped us to significantly improve the quality of our data analysis and presentation. In the revised version of this paper, we have tried, as much as possible, to take into account the comments and suggestions presented in the review.
Reviewer 3 Report
The quality of the manuscript has been improved visibly through author‘s modification. So the manuscript is acceptable to publish now. One minor suggestion for the author is that some explanation in Fig.8 could be changed to English if it is possible.
Author Response
We thank the reviewer for his in-depth analysis of our manuscript and his comments and suggestions, which helped us to significantly improve the quality of our data analysis and presentation. In the revised version of this paper, we have tried, as much as possible, to take into account the comments and suggestions presented in the review.
Below are the responses to the comments
Comments for Author:
- The quality of the manuscript has been improved visibly through author‘s modification. So the manuscript is acceptable to publish now. One minor suggestion for the author is that some explanation in Fig.8 could be changed to English if it is possible.
Response: We have made a change in Figures 8.

Reviewer 4 Report
All collection are incorporated in revised version
Author Response

(The authors gave the same response as above.)
